# The Role of Cancer Stem Cells and Their Extracellular Vesicles in the Modulation of the Antitumor Immunity

**DOI:** 10.3390/ijms24010395

**Published:** 2022-12-26

**Authors:** Daria S. Chulpanova, Albert A. Rizvanov, Valeriya V. Solovyeva

**Affiliations:** Institute of Fundamental Medicine and Biology, Kazan Federal University, Kazan 420008, Russia

**Keywords:** cancer stem cells, immune system, extracellular vesicles, cancer immune surveillance

## Abstract

Cancer stem cells (CSCs) are a population of tumor cells that share similar properties to normal stem cells. CSCs are able to promote tumor progression and recurrence due to their resistance to chemotherapy and ability to stimulate angiogenesis and differentiate into non-CSCs. Cancer stem cells can also create a significant immunosuppressive environment around themselves by suppressing the activity of effector immune cells and recruiting cells that support tumor escape from immune response. The immunosuppressive effect of CSCs can be mediated by receptors located on their surface, as well as by secreted molecules, which transfer immunosuppressive signals to the cells of tumor microenvironment. In this article, the ability of CSCs to regulate the antitumor immune response and a contribution of CSC-derived EVs into the avoidance of the immune response are discussed.

## 1. Introduction

Transformed cells in tumors are often a mixture of genetically heterogeneous cells, changes in the genome of which can occur both sequentially and in parallel. Such differences in the genotype and, consequently, in the properties of tumor cells inside one tumor can create significant difficulties when selecting an effective treatment strategy [1]. However, in addition to genetically different cells, non-genetic heterogeneous populations, tumor stem cells (CSCs) and non-CSCs in particular, have been found in tumor tissue [2].

According to the current data, CSCs represent a small population of the cells in tumor tissue, which are characterized by properties different from other non-CSC tumor cells. CSCs are able to self-renew and differentiate into heterogeneous cancer cell lines. According to one of the currently existing models of tumor formation, cancer stem cells differentiate into non-CSCs, thereby ensuring the emergence, progression and recurrence of the tumor, as well as the formation of metastases [3].

Cancer stem cells and normal stem cells have similar cell surface markers, and by analyzing them they can be distinguished from the general pool of tumor cells. Markers may vary depending on the type of tumor, but most CSCs carry CD133, CD44, CD90, CD34, aldehyde dehydrogenase 1 (ALDH)-1, epithelial cell adhesion molecule (EpCAM) and other stem cell markers on their surface [4]. Markers for some common cancer types are presented in Table 1. In addition, to be identified as CSCs, cells must be able to form tumorspheres in vitro and tumors in immunocompromised mice in vivo [5,6].

CSCs are also identified in articles as tumor initiating cells, which perfectly reflects the ability of these cells to form metastatic foci [38]. Due to the high level of proteolytic molecules, CSCs are able to remodel extracellular matrix and migrate from the primary tumor focus to another tissue through cardiovascular or lymphatic system [39,40]. Upon reaching the metastatic niche, CSCs can stimulate the formation of new vessels by increasing the expression level of vascular endothelial growth factor (VEGF) [41], as well as transform on their own into vascular progenitor cells from which new vessels are formed [42,43].

Many studies also indicate that CSCs mediate resistance to a wide range of chemotherapeutic drugs, including cisplatin [44], paclitaxel [45], doxorubicin [46], etc. This resistance is partially mediated by the fact that CSCs are located in a typical stem cell niche, providing protection of cells from the cytotoxic effects of drugs [47]. Moreover, CSCs also have many mechanisms that ensure the efflux of drugs from the cells [48], inactivation of drugs inside the CSCs using the mechanisms of normal stem cells [49] or due to overexpression or overactivation of DNA repair proteins [50]. In addition, CSCs mostly exist in the nondividing phase of the cell cycle, which allows them to avoid cytotoxic effects of chemotherapeutic drugs that target rapidly proliferating cells [51]. In addition, CSCs have mechanisms that allow them to regulate immune cells, evade immune response and create an immunosuppressive environment in the tumor niche [52].

Pro-tumor properties of CSCs are mediated by the interaction of tumor cells with the surrounding cells and non-CSCs through various signaling mechanisms [52]. An important part of the communication mechanism between CSCs and the tumor microenvironment (TME) is extracellular vesicles (EVs), small membrane structures formed by the cells as a part of normal cellular metabolism. EVs are able to effectively transfer signaling molecules of the parental cells to the neighboring cells, thereby mediating the regulation of their properties, including metastasis, stimulation of angiogenesis and regulation of the immune response [53]. In this review, we discuss how CSCs, as well as their EVs, regulate the antitumor immune response, which allows them to evade immune surveillance, as well as to modulate the tumor’s immune microenvironment.

## 2. Regulation of the Immune Response by CSCs

Normal human stem cells have low immunogenicity, so immune response evasion is another fundamental property of CSCs, which share many of the features of stem cells [54]. Due to these capacities, CSCs are able to effectively evade the antitumor immune response. The main immunosuppressive effects of CSCs on the immune cells are reflected in Figure 1.

In order to recognize transformed cells, effector CD8^+^ T-cells bind major histocompatibility complex (MHC)-I receptors, which present antigens on the surface of tumor cells, thereby triggering a complex sequential process of tumor cell killing [55]. One of the possible mechanisms that CSCs use to avoid immune surveillance is a disruption of the mechanisms of antigen processing and presentation. It has been shown that melanoma CSCs organized into spheroids carried a limited number of MHC-I molecules on their surface compared to adherent melanoma cells, which were almost completely positive for MHC-I [56]. Neurospheres formed by glioblastoma CSCs also showed low expression of MHC-I, as well as the absence of MHC-II expression [57]. It is important to note that the low level of MHC-I molecules on the cell membrane is an important property of normal stem cells, which determines their low immunogenicity [58,59]. Probably, the lack of MHC expression is also associated with the disturbances in antigen processing in the tumor cells. This is confirmed by the fact that key molecules involved in antigen processing including low-molecular-weight protein (LMP), transporter associated with antigen processing (TAP) and beta-macroglobulin were downregulated in glioblastoma CSCs [57].

### 2.1. T-Cells

The effector activity of T-cells is an important factor in the process of tumor elimination. Defects in T-cell activation both in the microenvironment and in circulation have been found in patients with various types of tumors [60]. In addition to the low expression of MHC-I on the surface of CSCs, the level of costimulatory molecules, such as CD80 and CD68, which are necessary for T-cell activation, was also reduced [61]. For example, glioblastoma CSCs expressed significantly restricted levels of the CD86 costimulatory molecule [62]. At the same time, CSCs expressed high levels of the immune checkpoint inhibitor molecule programmed death-ligand 1 (PD-L1), which led to a decrease in tumor infiltration of CD8^+^ T-cells in prostate cancer patients [63]. Such ligands as PD-L1 suppress the activation of T-cells, depriving their ability to effectively kill tumor cells. CSCs also express other immune checkpoint molecules CD276, cytotoxic T-lymphocyte-associated antigen 4 (CTLA4) and galectin-3 (Gal-3) [64,65]. A blockade of CD276 (B7-H3) checkpoint molecules on the surface of human head and neck squamous cell carcinoma (HNSCC) CSCs provided T-cell-dependent elimination of these cells, inhibiting tumor growth and metastasis formation in the HNSCC mouse model in vivo [66]. At the same time, Gal-3, which has immunosuppressive properties, was expressed in prostate CSCs made them less susceptible to apoptosis in vitro [67].

In addition, CSCs also secrete cytokines and growth factors that can modulate T-cell antitumor immune response. It has been shown that breast CSCs secrete a relatively high level of transforming growth factor-β (TGF-β) compared to non-CSCs [68]. Secreted from gastric CSCs, TGF-β induced differentiation of naive T-cells into regulatory T-cells (Tregs) [69,70], which support escape from immune response via inhibiting the activation and differentiation of CD4^+^ helper T-cells and CD8^+^ cytotoxic T-cells [71].

### 2.2. Dendritic Cells

In addition to the low expression of MHC-I/II molecules, the relatively low expression of tumor-specific antigens (TAA) on the surface of cancer stem cells compared to non-CSCs also contributes to the avoidance of immune surveillance [72]. Normally, antigen-presenting cells (APCs) (such as dendritic cells (DCs) and macrophages) process tumor cell antigens and present them on their surface for subsequent T-cell activation [73]. However, CSCs carry on their surface-reduced TAA number with predominant stemness markers CD133 [74], CD44 [75], EpCAM [76], etc., which are not recognized by APCs as foreign antigens. Along with this, CSCs can also directly regulate the properties of DCs. For example, secreted by hepatocellular carcinoma CSCs, TGF-β also led to the formation of tolerogenic DCs in the TME and the development of DC-mediated immunotolerance in an HCC mouse tumor model in vivo [77]. In turn, regulatory DCs, such as follicular DCs, maintained tumor growth and chemoresistance through the (C-X-C motif) ligand 12/chemokine (C-X-C motif) receptor 4 (CXCL12/CXCR4) axis [78].

### 2.3. Macrophages

CSCs also form close interdependent communication with another type of antigen-processing cells, macrophages [79]. Numerous data indicate that CSCs recruit both mature macrophages and their monocyte precursors into TME through the secretion of multiple chemokines. For example, Sox2^+^ breast cancer cells attracted macrophages to the TME by secreting (C-C motif) chemokine ligand 3 (CCL3) and intercellular adhesion molecule 1 (ICAM-1) chemokines [80]. It has also been shown that glioblastoma CSCs secrete periostin (POSTN), which binds to integrin αvβ3 and recruits predominantly monocyte-derived macrophages from peripheral blood in glioblastoma mouse models in vivo [81]. Glioblastoma CSCs also secreted CCL5, VEGF-A and neurotensin as chemoattractants for macrophages [82]. Most of these studies analyzed the ability of CSCs to recruit tumor-associated macrophages (TAMs), which exhibit the immunosuppressive properties of M2-like macrophages [83]. TAMs play an important role in the maintaining of tumor progression, and their recruitment to TME via CSC-secreted chemokines can make a significant contribution to tumor progression. Due to the immunosuppressive properties typical of stem cells, CSCs are also able to induce polarization of pro-tumorigenic macrophages into immunosuppressive M2-like TAMs [84]. For example, glioblastoma CSCs isolated from patient tissue triggered polarization of macrophages/microglia towards the immunosuppressive M2 phenotype in vitro [85]. Macrophages recruited into the TME, in turn, form a niche around CSCs and actively support the viability of tumor stem cells. For example, TAM-associated secretion of milk fat globule-epidermal growth factor-VIII (MFG-E8) activated signal transducer and transcription pathway activator-3 (Stat3) in non-small cell lung cancer CSCs, which enhanced their drug resistance in a mouse model in vivo [86]. High levels of interleukin 6 (IL)-6 generated by TAMs stimulate the proliferation of hepatocellular carcinoma CSCs in vitro [87]. In the studies dedicated to non-small cell lung cancer, IL-10 upregulated Janus kinase 1 (JAK1) signaling, which also supported tumor growth [88]. It is important to note that many cytokines secreted by TAMs trigger signaling pathways in stem cells aimed at maintaining their stemness. Binding of TAM surface ligands Thy1 and EphA4 with EphA4 receptor of breast CSCs activated Src and NF-κB pathways [89], which in turn reinforced the secretion IL-6, IL-8 and granulocyte–macrophage-colony stimulating factor (GM-CSF), providing the self-renewal of tumor cells in vitro [90]. In addition, TAMs also contribute to CSC immune response avoidance. Macrophages were able to induce the overexpression of the CD47 ligand on the surface of various tumor cells, which binds to the signal regulatory protein α (SIRPα) on the surface of phagocytic cells, inhibiting their phagocytic activity [91,92].

### 2.4. Natural Killers

The effect of CSCs on natural killer (NK) cells is not as unambiguous as on other types of immune cells. On the one hand, since MHC-I expression is reduced on the surface of tumor stem cells, their ability to induce inhibitory signals in NK cells is restricted [93]. Effective elimination of these cells by activated NK cells has been shown for many types of CSCs, including glioblastoma, melanoma and colon cancer cells [94,95,96]. In addition, some stem cells express typical CSC markers on their surface, which activate the cytotoxic functions of NK cells. For example, NK cells preferentially induced death of CD24^+^CD44^+^CD133^+^ALDH^bright^ CSCs compared to corresponding non-CSC populations, probably by binding NKG2D-activating receptors with its ligands MHC-I chain-related protein A and B (MICA/B) or death receptors (DRs) Fas and DR5 on the membrane of CSCs [97]. It is interesting to note that another type of effector cells whose activity depends on the expression of CD1a on the cell surface, invariant natural killer T (iNKT)-cells, also showed moderate cytotoxic activity against colorectal cancer stem cells [98].

On the other hand, CSCs also have a significant number of mechanisms capable of suppressing the NK-mediated immune response. For example, breast CSCs downregulate the expression of MICA and MICB, which stimulate NK cell activation. Such suppression of activation receptors caused by aberrantly expressed oncogenic miR20a provided breast CSC resistance to the cytotoxic activity of NK cells in vitro [99]. Moreover, glioma CSCs expressed low levels of the ligands for NKp30, NKp44, NKp46 and NKG2D, the expression of which was elevated when tumor cells were treated with interferon (IFN)-γ, allowing NK cells to induce the death of CSCs in vitro [100]. Some stem cells can also enter a dormant state due to autocrine stimulation of the Wnt signaling pathway by the Dickkopf-1 (DKK1) inhibitor, which suppresses stem cell proliferation. The transition to the resting state leads to a decrease in the expression of cell surface UL16-binding protein (ULBP) activator of NK cell-mediated cytotoxicity, thereby providing evasion from NK cell cytotoxic activity [101].

## 3. Extracellular Vesicle-Mediated Suppression of Immune Response

Extracellular vesicles are released by all the tumor-forming cells, including CSCs. EVs can be divided into three groups depending on their biogenesis: exosomes, microvesicles (MVs) and apoptotic bodies (ABs). Exosomes are small spherical structures 40–100 nm in diameter, which are formed during endocytosis inside multivesicular bodies [53]. MVs have a diameter of 100–1000 nm and are released by budding directly from the cell membrane [102]. Unlike exosomes and MVs, which are continuously produced by the cells, ABs are formed as part of fragmentation of cells undergoing apoptosis [103]. Extracellular vesicles are capable of transporting their cargo of parental cells represented by various proteins and nucleic acids such as DNA, messenger RNA (mRNA), microRNAs (miRNAs) and other noncoding RNA (ncRNA) [104].

CSC-derived extracellular vesicles carry specific molecules capable of recruiting tumor-supporting cells as well as maintaining tumor heterogeneity [105]. For example, CSC-derived EVs carry parental stemness markers such as CD44v6, tetraspanin 8 and neurogenic locus notch homolog protein 1 (Notch1), which typically activate for CSCs signaling pathways in non-CSCs [106,107,108]. Moreover, exosomes derived from clear cell renal cell carcinoma (CCCRC) stem cells accelerate the metastasis-promoting epithelial–mesenchymal transition (EMT) of non-CSCs by miR-19b-3p transfer, downregulating phosphatase and tensin homolog deleted on chromosome 10 (PTEN) expression in vitro [109]. A similar metastasis-supporting effect has been shown for lung cancer exosome-carried miR-210-3p, which upregulated the expression levels of N-cadherin, vimentin, matrix metallopeptidase (MMP)-9 and MMP-1, and downregulated E-cadherin expression in lung non-CSCs [110]. Moreover, exosomes from CSCs carry a large number of different molecules that can stimulate angiogenesis, including VEGF, MMP-2, MMP-9, miR-21, etc., [111,112]. The effect of CSC-derived EVs also extends to the regulation of the tumor’s immune environment. The suppressing effects of extracellular vesicle-transferred molecules on the immune cells are reflected in Figure 2.

### 3.1. T-Cells

As mentioned above, a large number of studies are dedicated to the evaluation of the immunosuppressive properties of glioblastoma CSCs [60]. Apparently, glioblastoma CSC-derived vesicles also contribute to the suppression of T-cell activity. Glioblastoma CSCs produce an extracellular matrix protein tenascin-C (TNC), which is packaged into exosomes and transported to the T-cells in both the TME and cardiovascular system. Exosomal TNC inhibited T-cell receptor (TCR) signaling through interaction with α5β1 and αvβ6 integrins and subsequent downregulation of the mTOR pathway in T-cells [113]. Another investigation of exosomes secreted by colorectal CSCs showed that exosomes contain a large amount of miRNA-146a-5p, and a high content of exosomal miR-146a expression in the serum of patients correlates with a decreased number of tumor-infiltrating CD8^+^ T-cells [114].

### 3.2. Neutrophils

The role of neutrophils in the tumor immunity remains controversial, since there are two contradictory populations, N1 and N2, which can support or suppress tumor progression, correspondingly. Neutrophils promote tumor growth, mainly by the stimulation of angiogenesis or suppression of T-killer-mediated antitumor response [61]. A previously mentioned article has also indicated that the high level of exosomal miR-146a in the serum of colorectal cancer patients is associated with the increased number of tumor-filtrating CD66^+^ neutrophils [114]. The tumor-supporting effect of CD11b^+^Ly6G^High^Ly6C^Low^ neutrophils recruited to the tumor by exosomes secreted by CSCs has been shown in a mouse model of colorectal cancer. Exosomal RNA induced IL-1β expression in neutrophils, which were then recruited in the TME by colorectal CSCs and supported tumor cell growth [115].

### 3.3. Macrophages

As described above, there are mutually supportive relationships between macrophages and CSCs. Some studies have shown that the effect of CSCs on macrophages is also mediated by secreted vesicles. Thus, monocytes were able to actively internalize exosomes of glioblastoma CSCs, which in turn caused active rearrangement of the monocyte cytoskeleton and their differentiation into M2-like macrophages expressing high levels of PD-L1 on their surface in vitro [116]. Conversely, M2-polarized macrophages induced and maintained stemness in non-CSCs, enhancing their ability to evade the immune response [117].

### 3.4. Dendritic Cells

Vesicles released by CD105^+^ CSCs inhibited the process of monocyte differentiation into DCs. The vesicles carried human leukocyte antigen (HLA)-G molecules, which suppressed the anti-tumor functions of NK cells, T-cells and DCs in vitro [118]. Monocytes, after the interaction with the renal CSC-derived EVs, retained the monocyte/macrophage marker CD14, while not upregulating CD1a, HLA-DR, activation markers CD83, CD80 and adhesion molecules α4 integrin and CD54, as well as failing to stimulate T-cell proliferation [119].

### 3.5. Myeloid-Derived Suppressor Cells

Previous studies have already shown that monocytes can actively internalize glioblastoma CSC-derived vesicles. In addition to triggering the differentiation of monocytes into M2-polarized macrophages, vesicles can also initiate the differentiation of the same cells into monocytic myeloid-derived suppressor cells (moMDSCs) that actively secrete arginase-1 and IL-10. The presence of moMDSCs in a population of tumor-associated lymphocytes led to a significant suppression of the CD3^+^ T-cell proliferation in vitro [120].

## 4. Conclusions

The CSC-based concept of tumor origin posits that cancer cells with stem properties are tumor initiators and can promote metastases and disease recurrence after conventional therapy. CSCs are weakly immunogenic cells and commonly do not carry MHC-I and MHC-II on their surface, which allows them to evade antitumor immune response. In addition, CSCs actively modulate their microenvironment, including immune system cells. CSCs suppress T-cell activation and also redirect their differentiation toward Tregs that suppress immune response against the tumor. In addition, they suppress the antigen presentation ability of DCs and also induce the polarization of macrophage into immunosuppressive M2-like TAMs.

Extracellular vesicles have recently received the status of one of the most important signal carriers for all the cell types. A small number of studies dedicated to the investigation of the effects of CSC-derived vesicles on cells of the immune system has shown that EVs, like the parental cells, suppress the activation of T-cells and DCs, increase the number of neutrophils with protumor properties in the TME, and are also able to trigger monocyte differentiation into M2-like macrophages or moMDSCs. However, the number of original research articles dedicated to the contribution of EVs to CSC-mediated suppression of the antitumor immune response is limited. Probably, part of the effects described for CSCs is determined precisely by the release of vesicles, and the modulation of their secretion can reduce the degree of the immune evasion. In addition, the vast majority of studies have focused on EVs secreted by glioblastoma CSCs. This is probably due to the fact that glioblastoma is characterized by a significant local and systemic immunosuppression, a low number of effector T-cells, and a large number of circulating immunosuppressive cells, such as Tregs and MDSCs [121]. However, the contribution of CSC-derived EVs to avoiding the immune response in other cancer types remains considerably unexplored.

Existing immunotherapy approaches directed to CSCs are largely based on genetic modification of T- or NK cell receptors in order to target them to CSCs, thus overcoming the escape of cells from immune surveillance [122,123]. For example, the efficacy of CD133-targeting chimeric antigen receptor T-cells has been investigated in a Phase I clinical trial in patients with hepatocellular carcinoma, pancreatic carcinoma, and colorectal carcinoma. Three of the 23 patients achieved partial remission and 14 patients had disease stabilization [122]. NK cells are able to relatively efficiently eliminate CSCs compared to the other effector cells. However, for the treatment of NK-resistant tumors, natural killer cells can be activated by IL-2 [124] or modified to target CSC receptors, such as the epidermal growth factor receptor (EGFR) [125]. In addition, vaccines based on DCs loaded with specific CSC antigens are currently under investigation [126]. For example, DCs loaded with gastrointestinal CSC mRNA can elicited tumor-specific T-cell immune response in vitro [127]. DCs loaded with Nanog homeobox (NANOG) peptide can potentially evoke strong T-cell-mediated anti-tumor response against ovarian cancer cells [128].

Understanding the contribution of CSC-derived extracellular vesicles to immune escape of the tumor should allow for the development of new immunotherapy protocols aimed at modulating the release of EVs by CSCs. Suppression of EV release by tumor cells can be achieved with clinically approved drugs such as amiloride, which blocks both biogenesis and macropinocytosis of EVs, as well as imipramine, calpeptin, manumycin A, etc., [129,130]. The antitumor effect of imipramine is currently under investigation in clinical trials in patients with ER-positive and triple-negative breast cancer (NCT03122444) and recurrent glioblastoma (NCT04863950). Blocking the release of vesicles can make possible achieving the sensitization of CSCs to the cytotoxic effect of immune cells, as well as increasing the effectiveness of the existing immunotherapy approaches described above. Therefore, future studies should focus on expanding our understanding of the effect of CSC-derived vesicles on the immune system’s cells and evaluating the outcome of EV biogenesis suppression on tumor immune status.

## Figures and Tables

**Figure 1 ijms-24-00395-f001:**
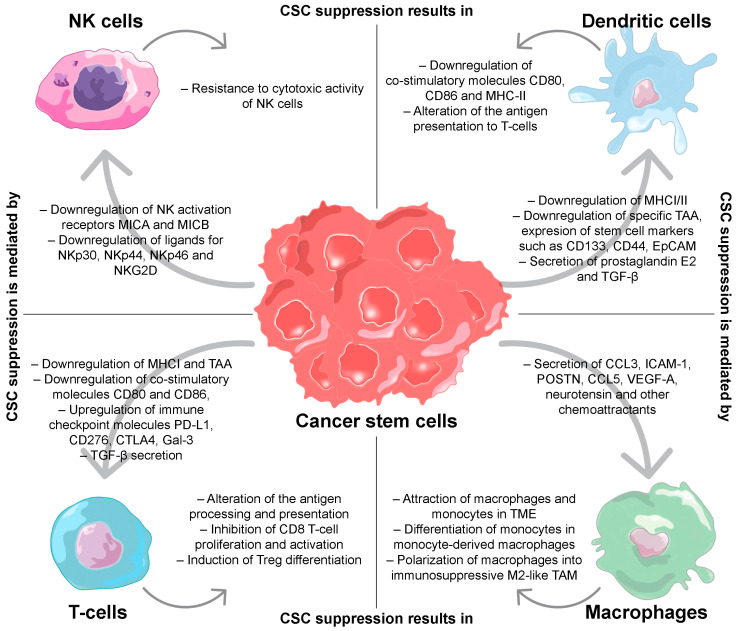
The influence of CSCs on the cells of the immune system is mediated by low immunogenicity of cancer stem cells (reduced expression of MHC-I/II, altered antigen processing and presentation, downregulation of TAA), suppression of the activation of effector T-cells and NK cells due to receptors located on the cancer stem cell surface, as well as secreted factors.

**Figure 2 ijms-24-00395-f002:**
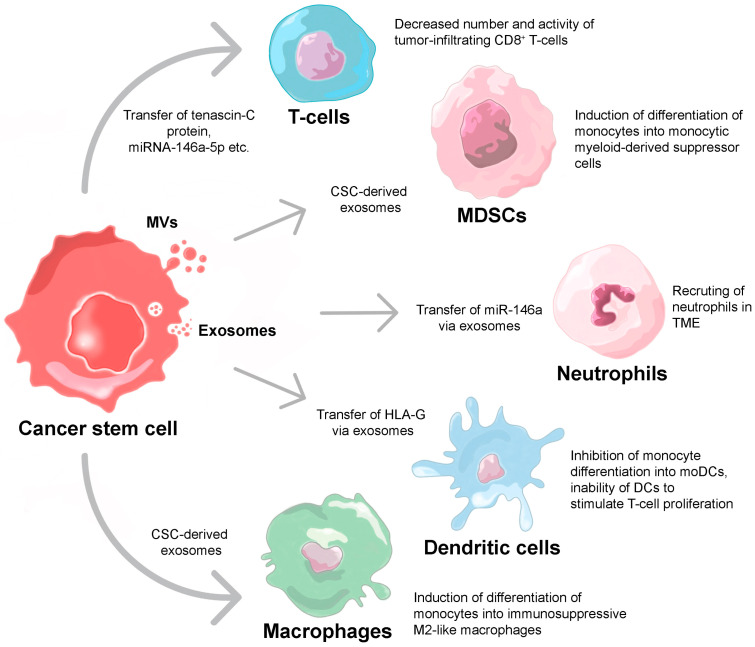
The influence of extracellular vesicle-transferred molecules on immune cells. Transfer of the specific proteins and mRNAs from parental CSCs via extracellular vesicles mediated suppression of cytotoxic T-cells and DCs as well as recruiting of tumor-supporting neutrophils, macrophages and MDSCs.

**Table 1 ijms-24-00395-t001:** Common CSC markers for various types of cancer.

Cancer Type	Membrane Markers	Intracellular Markers	Markers Commonly Used for Identification	References
Lung cancer	CD44, CD87, CD90, CD133, CD166, EpCAM, ALDH1	Nanog, Oct-3/4	CD87^+^CD133^+^	[7,8,9]
Breast cancer	CD44, CD61, CD90, CD133, CXCR4, Lgr5, ProC-R, ALDH1	Notch, Oct-3/4, Sox2	CD44^+^CD24^−^ALDH1^+^	[10,11,12,13,14,15,16,17]
Gastric cancer	CD24, CD44, CD90, CD133, Lgr5, ALDH1, CXCR4, ABC, Lrig1	DOCK6, Mist1, MSI-1, Nanog, Oct-3/4, Sox2	CD44^+^CD133^+^ALDH1^+^	[18,19]
Liver cancer	CD24, CD44, CD90, CD133, ALDH1, EpCAM	Nanog, Notch, Oct-3/4, Sox2	CD90^+^CD133^+^EpCAM^+^	[20,21,22,23]
Colorectal cancer	CD24, CD44, CD133, CD166, EpCAM, Lgr5	ALDH, Nanog, Oct-3/4, Sox2	CD24^+^CD44^+^CD133^+^	[24,25]
Acute myeloid leukemia	CD33, CD34, CD123, CLL-1, TIM3	ALDH, Nanog, Oct-3/4, Sox2	CD34^+^CD38^-^	[26,27]
Glioblastoma	CD44, CD70, CD133, ALDH1	Nanog, Oct-3/4, Sox2, Nestin	CD44^+^CD133^+^Nestin^+^	[28,29,30,31]
Head and neck squamous cell carcinoma	CD44, CD133, ALDH1	Nanog, Oct-3/4, Sox2, Bmi-1	CD44^+^CD133^+^ALDH1^+^	[32,33,34]
Melanoma	CD20, CD133, CD271, ALDH1, ABCB1/5, ABCG2	Nanog, Oct-3/4	CD133^+^CD271^+^ABCB5^+^	[35,36,37]

EpCAM: epithelial cell adhesion molecule; Lgr5: leucine-rich repeat-containing G-protein-coupled receptor 5; ALDH1: aldehyde dehydrogenase 1; CXCR4: C-X-C chemokine receptor type 4; ABC: ATP-binding cassette subfamily; Lrig1: leucine-rich repeats and immunoglobulin-like domains protein 1; DOCK6: dedicator of cytokinesis 6; Mist1: muscle, intestine and stomach expression 1; MSI-1: Musashi RNA-binding protein 1; Nanog: nanog homeobox; Oct-3/4: octamer-binding transcription factor 3/4; Sox2: sex-determining region Y-box 2; ProC-R: endothelial protein C receptor, CLL-1: C-type lectin-like molecule-1; TIM3: T-cell immunoglobulin and mucin domain 3; ABCB1/5: ATP-binding cassette subfamily b member 1/5; ABCG2: ATP-binding cassette sub-family G member 2, Bmi-1: B-cell-specific Moloney murine leukemia virus integration site 1.

## Data Availability

Not applicable.

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
