# Peer review of "The Role of Cancer Stem Cells and Their Extracellular Vesicles in the Modulation of the Antitumor Immunity"

_ijms, 2022, doi:10.3390/ijms24010395_

Round 1
Reviewer 1 Report
Attached

Author Response
Thank you for your comments which have helped us to improve our manuscript. We have corrected your comments point by point within the manuscript accordingly (your comments are in bold text and our responses are in ordinary type):
1) Abstract needs to include in the last sentence a description of novelty in this work. Abstract statement “However, it is not clear if extracellular 14 vesicles (EVs) secreted by CSCs make a significant contribution into the avoidance of the immune response” contradicts the supporting evidence in the review and Conclusions. Please modify.
It was rephrased in order to better reflect the data described in the article (Page 1, Lines 13-17).
2) Lane 191: However, on the other hand : Double negative, delete one
It was corrected.
3) Reference in several areas to “some CsC” or “CsC” without mentioning the tumor type leaves the impression that this is a general property of CsC. For example, lanes 195 and 199. Because of great heterogeneity in CsC population in different tumor types, each example should state what tumor was used. Lanes 195, 199, 244, etc…
We have checked throughout the entire manuscript and added appropriate information about tumor types (new information is highlighted in yellow in the text).
4) Remove one of )) from Lane 214, refrease the examples to : such as DNA, etc
It was corrected in accordance with your comment (Page 6, Lines 215-216).
5) Lane 284: “The CSC-based concept of tumor origin believes”, please modify omitting the word believes
It was corrected (Page 8, Line 287).
6) Lane 237: The statement “The effector activity of T-cells is an important factor in the process of tumor elimination. Defects in T-cell activation both in the microenvironment and in circulation have been found in patients with various types of tumors, including glioblastoma [111]” This statement belongs to the first portion of T cell function, 2.1
We have moved this information to Section 2.1 (Page 4, Lines 107-109).
7) Abbreviate CSC and EV consistently, Lane 245: “Another investigation of exosomes secreted by colorectal cancer stem cells showed…” Lance 215: “ CSC-derived extracellular vesicles:
We have checked throughout the whole manuscript and added appropriate abbreviations where it was necessary.
8) Lane 216: Statement needs to be supported by references: “maintaining tumor heterogeneity.”
The appropriate reference was added (Page 6, Line 218).
9) Lane 286: …”change “that” to “and”
It was corrected.
10) Lane 299: “obviously insufficient” is an overstatement, please modify
It was corrected (Page 8, Lines 301-303).
11) Lane 121: Unprecise reference 126, need to specify what model was used: in vivo, in vitro T cell response? Also in other references, response should be defined as in vitro assay of immune modulation, which cell type, clinical study, etc.
We replace the reference 126 with the appropriate one (Page 9, Lines 321-324). We have also checked the whole manuscript and added appropriate information about cell type, study type (new information is highlighted in yellow in the text).
Reviewer 2 Report
This is a well written manuscript and appears to refence most of the pertinent published papers relevant to the topic.
Author Response
Thank you for your comment.
Reviewer 3 Report
Chulpanova et. al. reviewed how the CSC and EVs regulate the antitumor immune response. The work is very well described. The reviewer has only one minor suggestion to improve the manuscript.
- Table 1: It is very difficult to see which markers are membrane, or intracellular. Please format the table to make it easy to see the columns.
Author Response
Reviewer 3.
Thank you for your comments. We have formatted the table so that the columns are better separated from each other (Page 2).